# Establishment of a *Coilia nasus* Gonadal Somatic Cell Line Capable of Sperm Induction *In Vitro*

**DOI:** 10.3390/biology11071049

**Published:** 2022-07-13

**Authors:** Yuting Kan, Ying Zhong, Muhammad Jawad, Xiao Chen, Dong Liu, Mingchun Ren, Gangchun Xu, Lang Gui, Mingyou Li

**Affiliations:** 1Key Laboratory of Integrated Rice-Fish Farming, Ministry of Agriculture and Rural Affairs, Shanghai Ocean University, Shanghai 201306, China; m200100064@st.shou.edu.cn (Y.K.); zhongy10@haid.com.cn (Y.Z.); jawadktk1293@gmail.com (M.J.); m190100049@st.shou.edu.cn (X.C.); dliu@shou.edu.cn (D.L.); 2Key Laboratory of Exploration and Utilization of Aquatic Genetic Resources, Ministry of Education, Shanghai Ocean University, Shanghai 201306, China; 3Key Laboratory of Microecological Resources and Utilization in Breeding Industry, Ministry of Agriculture and Rural Affairs, Guangzhou 511400, China; 4Key Laboratory of Freshwater Fisheries and Germplasm Resources Utilization, Ministry of Agriculture, Freshwater Fisheries Research Center, Chinese Academy of Fishery Sciences, Wuxi 214081, China; renmc@ffrc.cn (M.R.); xugc@ffrc.cn (G.X.)

**Keywords:** germplasm conservation, gonadal cell culture, spermatogonial stem cells, differentiation, spermatogenesis, spermatids

## Abstract

**Simple Summary:**

The induced differentiation of sperm cells *in vitro* is a key technology for germplasm conservation. *Coilia nasus* is one of the “Three Delicious Species in the Yangtze River”, its germplasm resources have declined significantly over the past few decades. To protect the declining of *Coilia nasus*, a gonadal somatic cell line from fresh gonadal tissues (*C*nGSC) of adult *Coilia nasus* was isolated that can be cultured and proliferated *in vitro* for a long time. Co-culture of *C*nGSC with a medaka spermatogonial stem cell (SG3) line led to the induction of sperm *in vitro*. Further experiments showed that it is the lysate of *C*nGSC that induced the differentiation of SG3 *in vitro*. Our findings will help to further study the conservation of germplasm resources of endangered fishes with important values.

**Abstract:**

*Coilia nasus* is an important economic anadromous migratory fish of the Yangtze River in China. In recent years, overfishing and the deterioration of the ecological environment almost led to the extinction of the wild resources of *C.nasus*. Thus, there is an urgent need to protect this endangered fish. Recently, cell lines derived from fish have proven a promising tool for studying important aspects of aquaculture. In this study, a stable *C. nasus* gonadal somatic cell line (CnCSC) was established and characterized. After over one year of cell culture (>80 passages), this cell line kept stable growth. RT-PCR results revealed that the CnGSC expressed some somatic cell markers such as *clu*, *fshr*, *hsd3β,* and *sox9b* instead of germ cell markers like *dazl*, *piwi,* and *vasa*. The strong phagocytic activity of CnGSC suggested that it contained a large number of Sertoli cells. Interestingly, CnGSC could induce medaka spermatogonial cells (SG3) to differentiate into elongated spermatids while co-cultured together. In conclusion, we established a *C. nasus* gonadal somatic cell line capable of sperm induction *in vitro*. This research provides scientific evidence for the long-term culture of a gonadal cell line from farmed fish, which would lay the foundation for exploring the regulatory mechanisms between germ cells and somatic cells in fish.

## 1. Introduction

Several fish species are facing extinction due to rapid population declines [1]. Research on fish germplasm resource conservation is increasingly important for the protection of endangered fish [2,3]. Cryopreservation of different cellular types (sperm, oocytes, embryos, early spermatogonia, and primordial germ cells or somatic cells) is a long-term germplasm conservation technique plays an important role in the reproductive practices of cultured aquatic species and assisted reproduction [1,4].

Fish cell line culture preservation is considered as an alternative aid to conserve fish germplasm [2]. The first established fish cell line is germ cell line of rainbow trout (*Oncorhynchus mykiss*), which raised the research on fish cell lines [5]. In general, fish cell lines have been developed globally using different types of tissue samples including kidney, brain, vertebra, and embryos [6,7,8,9]. Gonadal cell lines rarely develop from fish species with long-term culture conditions [10]. Fish gonadal cells are divided into germ cells and somatic cells. The testicular somatic cells are mainly composed of Sertoli cells and Leydig cells, of which the Sertoli cells play an important role in spermatogenesis as the survival and development of germ cells depend on their continuous and close contact with Sertoli cells [11].

Specifically, spermatogenesis in lower vertebrates can proceed *in vitro*, and sperm production in primary testicular culture has been reported for several fish species [11]. In addition, spermatogenesis induction *in vitro* requires a complicated and special cell culture system. In Teleostei, Sertoli cells delineate germ cells and promote their growth, meiosis, and gamete production [12]. In half-smooth tongue sole (*Cynoglossus semilaevis*), a testicular somatic cell line was established [13]. In yellow croaker (*Larimichthys crocea*), both ovary and testis somatic cell lines were obtained [14]. In zebrafish (*Danio rerio*), during 15 days of coculture on a feeder layer of Sertoli-like cells, dissociated testicular cells produced flagellated sperm [15]. In medaka (*Oryzias latipes*), long-term self-renewal cell division and differentiation potential of the SG3 cell line was established during 2 years in culture, showing spermatogonial features. Then, coculture with rainbow trout gonadal somatic cells induced motile sperm *in vitro* [16].

Spermatogenesis is a complex gonadal function in which spermatogonial stem cells continuously proliferate and differentiate from cells to form sperm after meiosis supported by somatic cells for more than one month in the body [17]. Naturally, spermatogenesis is regulated by a variety of intrinsic and extrinsic signals. Moreover, intrinsic factors include several important germ cell marker genes, such as *dazl*, *dnd*, *piwi,* and *vasa* [18,19,20], deletion of any one of these will lead to infertility. Meanwhile, extrinsic signals cover various hormones and growth factors, which adjust the development and differentiation of germ cells by regulating the activity and structure of Sertoli cells and the production of sex steroids [21,22]. Therefore, gonadal somatic cells are closely related to germ cells, which are essential for testis development and spermatogenesis of fish [23].

*Coilia nasus* is a significant fish species owing to its nutritional value and delicate flavour in the Yangtze River of China. It is considered a delicacy ranking first among the “Three Delicious Species in the Yangtze River”, [24]. Because the catch yield has been declining yearly, *C. nasus* is facing a survival crisis and has been listed on the National Key Protective Species List [25]. Research should be conducted on its reproductive biology to restore the diversity of germplasm resources of *C. nasus*. This paper set out to address this research gap. Recently, we have cloned germ cell genes of *C. nasus* [26]. In the present study, we established a gonadal somatic cell line of *C. nasus* called CnGSC and investigated the somatic cells capable of sperm induction. It was hypothesized that spermatogenesis *in vitro* would be associated with the interactions between germ cells and somatic cells in fish.

## 2. Materials and Methods

### 2.1. Fish and Cell Culture

*Coilia nasus* of 1.5 years old were harvested from Yixing, an experimental base of the Freshwater Fisheries Research Center of the Chinese Academy of Fishery Sciences. Primary cell culture was performed as described by [27]. Summarily, the fish were disinfected with 75% ethanol and incubated on ice for 10 min. The gonad was dissected and washed three times with phosphate-buffered saline (PBS) containing antibiotics (penicillin, 1000 IU mL^−1^; streptomycin, 1000 μg mL^−1^). Then, the tissue was transferred to a 1.5 mL EP tube placed on ice containing 500 μL of cold 0.25% trypsin–EDTA and cut into small pieces properly with scissors. Single fish testicular pieces were incubated in 1 mL cold trypsin–EDTA for 4 h on ice, then for 30 min at 28 °C. The pieces sank to the bottom. Without disrupting the pieces, trypsin–EDTA was aspirated, leaving 200 μL. Cell culture media was added to the tube and the cells were pipetted out. The cell suspension was transferred to a 0.1% gelatin-coated 12-well plate and cultured at 28 °C. After that, the medium was refreshed according to the cell growth conditions, and the cells were sub-cultured at a split ratio of 1:2. Cell clonal expansion and sub-culture were carried out as previously described [9,28]. The cells were sub-cultured every 3 to 4 days from the 30th passage onwards. The study was conducted in accordance with the Declaration of Helsinki and approved by the Shanghai Ocean University Animal Care and Use Committee with approval number SHOU-2021-118.

### 2.2. Cryopreservation and Recovery

Cryopreservation and recovery were performed as described previously with minor modifications [27]. For cryostorage, cells at 90% confluence were used. CnGSC cells digested by 0.25% trypsin–EDTA were resuspended in medium and centrifuged at 1200× *g* for 3 min. The cell pellet was resuspended in a cold medium (4 °C), containing 20% FBS, 10% dimethyl sulfoxide (DMSO), and 70% MEM media at a density of 5 × 10^6^ cells/mL. The cell suspensions with a density of 5–6 × 10^6^ cells/mL were transferred into a 1.8 mL sterile plastic vail. Then, the tubes were kept in a Nalgene Mr. Frosty Freezing Container, incubated for 4 h at −80 °C, and transferred into liquid nitrogen for cryostorage. In order to resuscitate the frozen cells, tubes were swiftly removed from the from the liquid nitrogen, thawed at 40 °C for 1 min, and then centrifuged at 1000 g for 4 min. The cells were resuspended into single-cell suspensions in fresh ESM4 and seeded on a 0.1% gelatin-coated cell culture flask.

### 2.3. Chromosome Analysis

Chromosome karyotype analysis was carried out as previously described [27]. In general, CSGC cells at passages 30, 50, and 70 were seeded into 25 cm^2^ culture flasks and incubated for 36 h at 28 °C. The cells were treated with colchicine at 0.1 μg/mL and incubated for 4 h. After that, the cells were treated with a 6 mL hypotonic solution of 75 mM KCl for 30 min. After a 4 h incubation period, the cells were treated for 30 min with a hypotonic solution of 0.075 M KCl. After that, 1 mL of fresh fixative (methanol 3; glacial acetic acid 1) were dropped above the cell suspension and then prefixed for 3 min. The cell pellets were fixed with cold fixative for 30 min after being centrifuged for 5 min at 1200× *g*. Cells were resuspended in 0.5 mL fresh fixative after centrifugation, then dropped and dispersed on cold glass slides by blowing. The slides were stained with 10% Giemsa (in 10 mM potassium phosphate, pH 6.8) for 30 min after air drying. Under slides were observed and photographed.

### 2.4. Gene Expression Analysis

Total RNA was obtained from the CnGSC cells using Trizol Reagent (GIBCO/BRL) and reverse transcription was performed with 1 μg of total RNA according to the cDNA synthesis kit protocols (Takara Bio, Kusatsu, Japan). PCR was performed using gene-specific primers (Table 1) for gonadal germ cells (*dazl*, *piwi*, *vasa*) and somatic cells (*clu*, *fshr*, *hsd3β*, *sox9b*). A PCR reaction system (20 μL) was operated for 28 (*β-actin*) and 38 cycles (the remainder) at 95 °C for 20 s, 60 °C for 30 s, and 72 °C for 1 min as described [26]. The amplified fragments were analyzed by agarose gel electrophoresis with a 50 bp or 1 kb DNA marker.

### 2.5. Phagocytosis Assay

The phagocytosis assay was carried out as to previously described [15]. CnGSC cells were incubated at 28 °C for 16–20 h in the medium containing polystyrene beads (1000:1, Sigma LB- 11, average diameter, 1.1 μm). The beads were washed three times with PBS. The morphology of CnGSC cells and the endocytosis of beads were examined under a light microscope. The presence of polystyrene beads in the cytoplasm of Sertoli cells indicates the cells’ phagocytic activity.

### 2.6. Cell Transfection

The medaka spermatogonial stem cell line, SG3, established previously was cultured at 28 °C in ESM4 [16]. SG3 cells at 70% confluence in 0.1% gelatin-coated 24-well plates were transfected with 1–2 μg pCVpr DNA using FuGENE^®^6 transfection reagent (Promega, Madison, WI, USA) to obtain a medaka spermatogonial stem cell line SG3, stably expressing a red fluorescent protein with instructions as mentioned previously [29]. pCVpr is a vector that expresses *pr*, a fusion protein combining puromycin acetyltransferase and red fluorescent protein (RFP) from CV, the human cytomegalovirus enhancer/promoter [30].

### 2.7. Preparation of CnGSC Lysate and Induced Differentiation

To prepare CnGSC lysate, cells at ~90% confluence were washed three times with PBS, digested with 0.25% trypsin–EDTA, and resuspended per 4 wells (about 2 × 10^7^ cells) in a 1 mL medium without basic fibroblast growth factor (bFGF) and medaka embryo extract (MEE). The cell suspension was filtered with a 0.22 μm filter after repeatedly freezing and thawing 3 times at −80 °C. At the same time, the culture supernatant of CnGSC cells as a control was collected in the same way. The medium was replaced by lysate or supernatant to induce SG3 cells *in vitro*.

### 2.8. Co-Culture

RFP-positive SG3 cells were kept at 28 °C in ESM4. After trypsinization, CnGSC single cells mixed with RFP-positive SG3 single cells were seeded into a 24-well plate at a ratio of 1:1. The coculture methods were as follows: culture at 28 °C and replace half of the medium every other day. Coculture details were measured as described previously [16], with modifications as described.

## 3. Results

### 3.1. Establishment of a Gonadal Somatic Cell Line

Fish gonadal tissue dissection experiments are shown in Figure 1. Dissociated cell masses were observed in isolated gonadal tissue after trypsin digestion (Figure 2A). After 5 days of culture at 28 °C, fibroblast-like cells grew to confluency in the primary culture (Figure 2B). With the passage of time and the increase in number, the morphology of the gonadal somatic cells gradually became single, appearing fibroblastic, uniform, and transparent in sub-culture (Figure 2C,D). In the ESM4 medium, the cells were sub-cultured every two days. To date, the cells have been sub-cultured to passage 78 and were still in a good proliferating state (Figure 2E). The *C. nasus* gonadal somatic cell line has been established and named CnGSC.

The CnGSC cells recovered at passage 38 proliferated to confluency in 3–4 days (Figure 2F). As a result, we successfully resuscitated cells in liquid nitrogen below −120 °C using 10% DMSO in the medium with a gradual freezing process.

### 3.2. Chromosome Analysis and Characterization of Somatic Cell Properties

Through karyotype analysis at passages 30, 50, and 70, CnGSC cells showed numerous abnormalities on the chromosomes. The results of the chromosome counts from 100 metaphase plates revealed that the number of chromosomes in CnGSC cells ranged from 44 to 79 (Figure 3A), with the majority (51%) having 49 chromosomes (Figure 3B). CnGSC cells separated from a sex-indeterminate adult gonad. The *C. nasus* shows a sexually distinct karyotype, with its ZZ-ZO chromosomes (2n(♀) = 47, 2n(♂) = 48) [31]. However, as heteroploidy and aneuploidy were observed in the CnGSC cell line, it cannot be determined that CnGSC cells are from the ovarian or testicular gonads.

To determine the identity of CnGSC cells, we checked the transcripts of some gonad expressed genes in CnGSC cells by RT-PCR. As shown in (Figure 4A), the expression of germ cells marker genes such as *dazl*, *piwi*, and *vasa* was not observed. On the contrary, high expression of marker genes of Sertoli cells including *clu* [32], *fshr* [33], and *sox9b* [34], and a marker gene of Leydig cells, *hsd3β* [35], was observed (Figure 4B). Thus, CnGSC cells contained somatic cells of both Sertoli cells and Leydig cells.

To further characterize CnGSC cells, a phagocytic latex bead assay was used to test the phagocytic activity based on the phagocytic properties of mammalian Sertoli cell lines [36]. The phenomenon of phagocytizing latex beads into the cytoplasm represents phagocytic activity (Figure 4D,E). Many of the cells showed phagocytic activity, which indicates the abundant existence of Sertoli cells.

### 3.3. Spermiogenesis

To check whether CnGSC cells have extensive ability to induce spermatogenesis, CnGSC cells were used to stimulate the differentiation of medaka spermatogonial stem cells (SG3). First, SG3 stably expressing red fluorescent protein was established for clear observation. Next, the CnGSC cells were co-cultured with the RFP-positive SG3 cells. Two days later, new, tightly compacted multicellular colonies with obvious boundaries of embryoid bodies were observed (Figure 5A). In consequence, the morphology of SG3 cells changed, and a transformation from the round spermatid to elongating spermatid with the newly formed flagellum was observed (Figure 5B–D). Therefore, CnGSC cells have the ability to induce SG3 to differentiate into sperm.

For more in-depth research, the lysate of CnGSC cells was used to induce SG3 cells. The results show that SG3 cells were induced by lysate 1–2 days later, and SG3 cells gradually formed round or elongated spermatids with different flagella lengths. On the third day, the presence of spermatid clusters with flagellum could be noticed. After 6–8 days, there were more spermatids with flagella. At the same time, the flagella gradually elongated and the sperm cell head shrank to a certain extent (Figure 6A–C). The culture supernatant of CnGSC cells as control could not give rise to obvious sperm (Figure 6D). Hence, the effective ingredients for inducing sperm production were inside CnGSC cells, not in the culture supernatant.

Furthermore, we also determined the meiotic gene expression of SG3 cells by using the mRNAs for *scp3* and *rec8b* to verify the process of meiosis. Synaptonemal complex protein 3 (Scp3), encoded by *scp3*, is a meiotic marker expressed in the nucleus of spermatocytes and is commonly used to trace the timing of gonadal differentiation in vertebrates [37]. *Rec8* is a meiotic cohesion gene expressed in the testis but not in the kidney or liver; more precisely, it is expressed in spermatocytes and spermatids but not in spermatogonia or other somatic cells in vertebrates [38,39]. Under normal culture conditions, a low level of mRNAs of both genes was present in SG3 cells. Under induction culture conditions, both genes’ expression levels were observed to be enhanced in SG3 cells (Figure 7).

## 4. Discussion

Fish cell lines are an extremely valuable tool because of their low cost and reproducibility and are widely used in cell biology, oncology, genomics, genetics, and resource conservation [1,40]. The present study reported the establishment of the *C.nasus* gonadal somatic cell line and differentiation of medaka SG3 into sperm, which verifies the effect of somatic cells on spermatogonial stem cells. Our study provides the first demonstration of the long-term culture of endangered fish gonadal somatic cells *in vitro*. Several experimental tests have confirmed the characteristics of the CnGSC cell line. First, CnGSC was of a gonadal origin, showing a fibroblast-like phenotype rather than round morphology in culture [16], and was provided with typical phagocytic activity [36]. Second, it expressed the marker genes (*clu*, *fshr*, *hsd3β*, and *sox9b*) of somatic cells but not germ cell marker genes (*dazl*, *piwi*, and *vasa*). Third, and most important, it could induce spermatogonial stem cells to differentiate from meiosis into sperm. An enhanced expression level of both meiotic marker genes (*rec8b* and *scp3*) in the spermatogenesis system was consistent with the result of inducing mouse spermatocytes *in vitro* [41]. Thus, the first endangered fish gonadal cell line (CnGSC) appeared to be derived from the fish testes and characterized the role of gonadal somatic cells, and expanded existing knowledge.

In our study, primary cultured cells were obtained using the mechanical dispersion method with enzyme digestion. The combination-design methods effectively separated cells and minimized cell damage. Fish cell culture is flexible to conditions such as subculture time, culture temperature range, and selection of culture medium. Therefore, the use of ESM4 medium, which was the previously reported medium for culturing medaka embryonic stem cells [27], and culture conditions can well maintain the mitotic ability of cells.

Results obtained from karyotype analysis showed numerous abnormalities in the chromosomes of CnGSC cells. The chromosome karyotype analysis of this cell line was shown to be aneuploid, with a chromosomal number of 49. Euploidy is an important parameter that reflects the characteristics of cell lines. Our result is inconsistent with a previously reported study on *C. nasus*. According to previous reports about *C. nasus*, the modal number of chromosomes is 2n(♀) = 47, 2n(♂) = 48 [31]. Aneuploidy occurs relatively frequently in diploid species of animals and plants due to the fusion of accidental 2n gametes and conventional 1n gametes, both produced by diploid individuals or caused by hybridization between diploid and tetraploid individuals, and they show various levels of fertility [42].

RT-PCR and phagocytosis tests were done to detect the identity of CnGSC cells. The results show that some Sertoli cell marker genes such as *clu, fshr*, and *sox9b* were expressed in CnGSC cells. In terms of the amount of phagocytically active cells, the cell line contains a considerable number of Sertoli cells. Sertoli cells are essential for spermatogenesis because they interact directly with SSCs. Some recent studies showed that Sertoli cells secreted small molecules to interact with germ cells by mediating cell connections [43,44]. Additionally, in vivo experiments also showed that transplanting mouse Sertoli cells into mouse testes with special treatment of Sertoli cells could promote the proliferation of mouse spermatogonia into the spermatocyte stage [45]. It seems that Sertoli cells could promote germ cell differentiation into meiosis and promote spermatogenesis. In addition, the phagocytic activity of Sertoli cells in CnGSC may help to eliminate apoptotic cells during spermatogenesis. This process is helpful to reveal the interaction between somatic cells and germ cells [46].

In our study, sperm cells were induced by a co-culture system and somatic cell lysate, and the process of spermatogenesis was simulated *in vitro*. Our results show that long-term self-renewal somatic cells or their lysates prompted round sperm morphogenesis by co-culture with medaka SG3 *in vitro.* In the research of *Anguilla japonica*, platelet-derived endothelial growth factor was found to be the key to inducing spermatogonial stem cell mitosis in a germ cell/somatic cell co-culture system [47], another culture condition is also a guarantee for the co-culture of germ cells/somatic cells to produce sperm cells [16]. In mammals, male germ cells could undergo the process of meiosis to differentiate into non-viable sperm by cell culture *in vitro* [48,49,50]. In mice (*Mus musculus*), microinjection of spermatids or sperm could produce healthy and fertile offspring. Neonatal testicular tissues were cryopreserved, and after thawing, showed complete spermatogenesis *in vitro* [50,51].

## 5. Conclusions

In summary, we successfully established a gonadal cell line from *C. nasus*, which was sufficient to induce sperm products by co-culturing C. *nasus* gonadal somatic cells with medaka SG3 cells. The role of gonadal somatic cells in the development of fish SSCs into sperm was briefly reviewed, providing strong evidence for long-term cell culture *in vitro* studies on sperm production. *In vitro* stimulation of the growing microenvironment of SSCs is an important component in spermatogonial stem cell development.

*In vitro* sperm production may simplify the production of endangered fish seedlings with high commercial value. Using the *in vitro* spermatogenesis protocol established in the current study would make it possible for sperm produced in Petri dishes to act as gametes to produce offspring of endangered species. In addition, the *in vitro* gonadal somatic cell culture system established in this study provides a basis for studying spermatogenesis, especially the interaction between somatic cells and spermatogonial stem cells. Our gonadal cell line can induce sperm production, it should be from the testes and its identification needs further study. In the present study, we did not obtain the spermatogonia stem cell line, maybe because the fish used were too old and contained very few spermatogonial stem cells, which prevented us from obtaining germline stem cells. In this case, it is necessary to get germline stem cells from the gonads of young fish.

## Figures and Tables

**Figure 1 biology-11-01049-f001:**
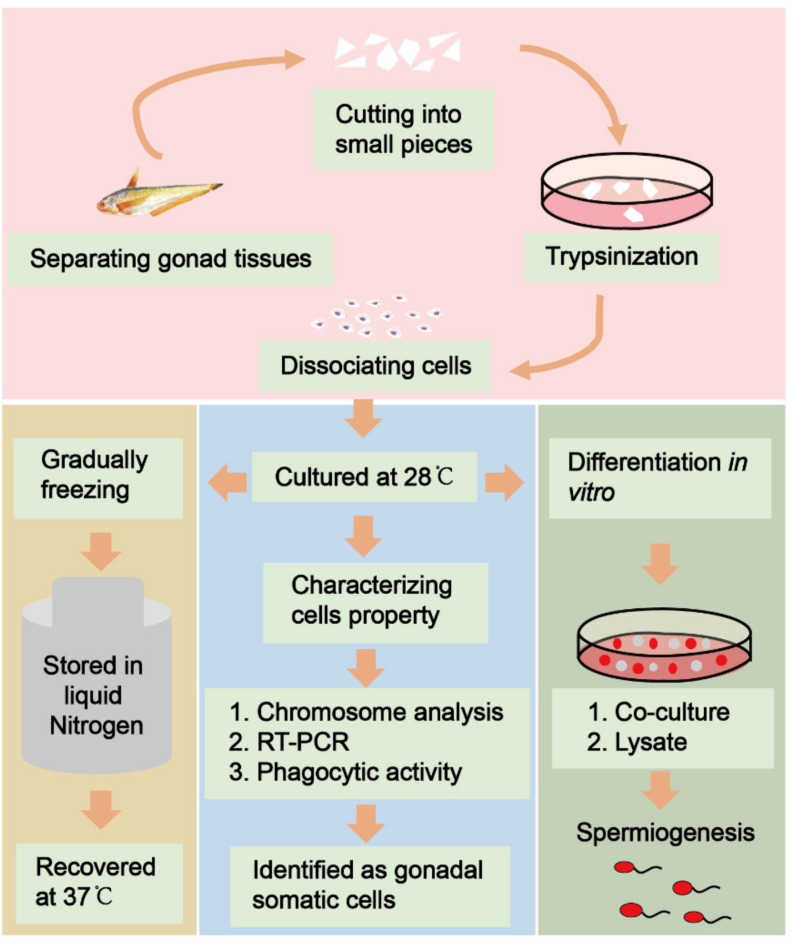
The flowchart of experiments. In total, 12 sex-indeterminate adult fish of 1.5 years old having a length of about 20–30 cm were used in this study. Live fish were placed on ice and executed. All gonadal tissues were dissected and washed three times with phosphate-buffered saline (PBS) containing antibiotics. They were cut into small pieces to dissociate cells by trypsinization. The stable cell lines were obtained by suitable culture and then frozen for storage. We verified that the cells were gonadal somatic cells by chromosome analysis, RT-PCR, and phagocytosis assay. Most importantly, spermatogonial stem cells were induced to differentiate into elongating spermatids by a co-culture system or gonadal somatic cell lysate culture conditions.

**Figure 2 biology-11-01049-f002:**
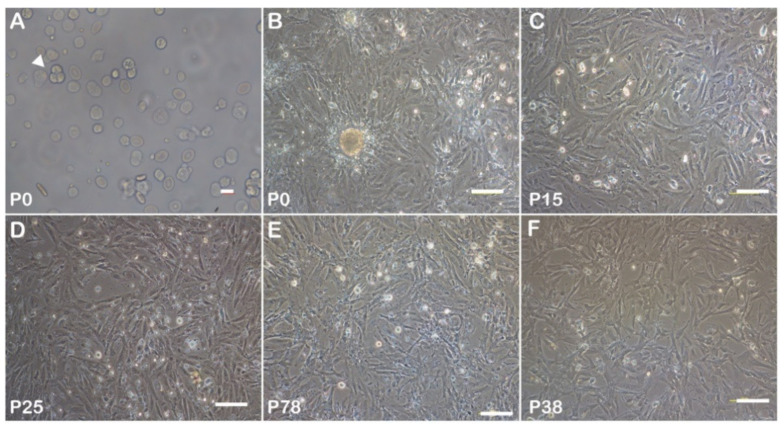
*Coilia nasus* gonad somatic (CnGSC) cells were cultured *in vitro*. (**A**) Cell mass was isolated from the gonads (arrow). (**B**) Primary culture of somatic cells for 5 days. (**C**) Morphology of somatic cells at passage 15 in sub-culture. (**D**) Sub-cultured CnGSC cells at passage 25. (**E**) Morphology of CnGSC cells at passage 78. (**F**) The monolayer of CnGSC cells (passage 38) recovered after three months. The cells retain their pre-cryopreservation morphology and proliferative ability. (Bars = 10 µm in (**A**); 100 µm in (**B**–**F**)).

**Figure 3 biology-11-01049-f003:**
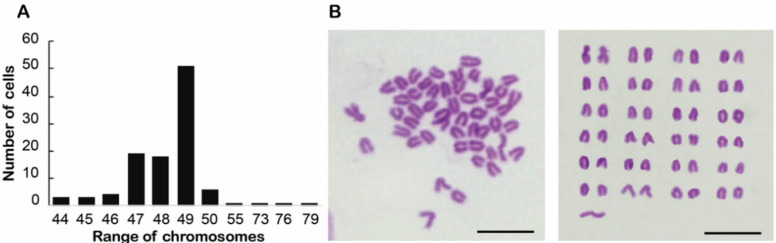
Chromosomes of CnGSC cells at passage 30, 50, and 70. (**A**). Analysis of chromosome distribution in cells. The main chromosome number was 49. (**B**) Metaphase chromosomes were micrographed after Giemsa staining. An example of a metaphase plate of CnGSC cells with hyperploid (49) chromosomes. (Bars = 10 µm in (**B**)).

**Figure 4 biology-11-01049-f004:**
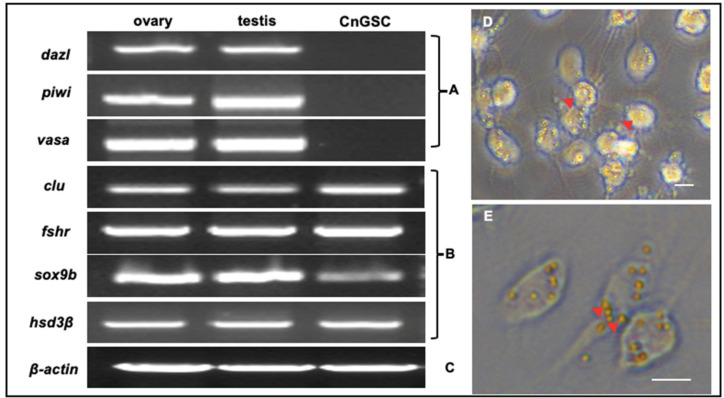
Characterization of CnGSC cells’ properties. (**A**–**C**) Gene expression pattern of CnGSC cells. (**A**) Absence of germ cell markers by RT-PCR with primers for *dazl*, *piwi,* and *vasa*. (**B**) Expression of somatic cell markers by RT-PCR of total RNA from gonad tissues and CnGSC cells. Sertoli cell markers: *clu, fshr, sox9b*; Leydig cell marker: *hsd3β*. (**C**) The *β-actin* gene was used as an internal control for RT-PCR. (**D**,**E**) Phagocytic activity. CnGSC cells showed phagocytic activity in terms of the uptake of latex beads. (Bars = 10 µm in (**D**,**E**)).

**Figure 5 biology-11-01049-f005:**
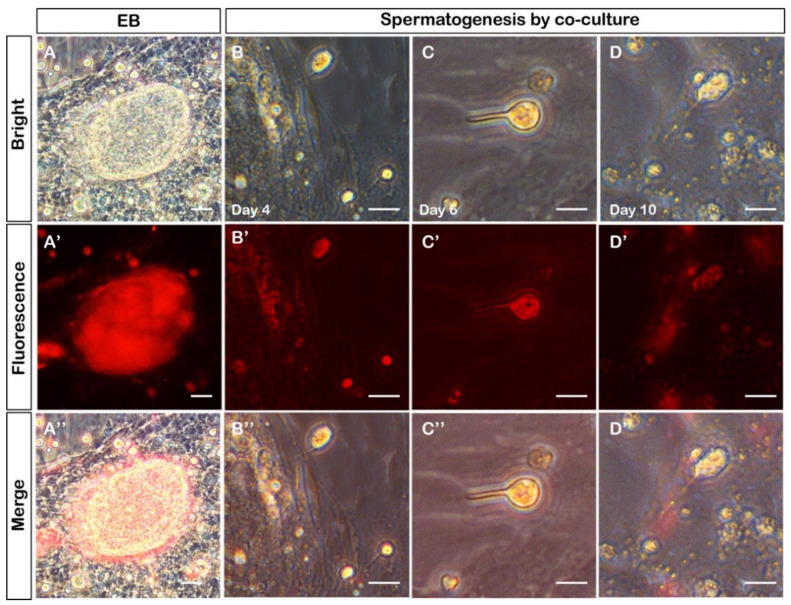
Sperm production *in vitro*. (**A**) Embryoid bodies were produced by co-culture of CnGSC cells and RFP-labelled SG3 cells. The large spherical structure is wrapped with SG3 cells and surrounded by somatic cells with a clear boundary. (**B**,**C**) SG3 cells differentiated into round spermatids with flagellum *in vitro* 4 days later. (**D**) The tail of sperm cells elongated 10 days later. (Bars = 100 µm in (**A**–**A”**); Bars = 10 µm in (**B**–**D”**)).

**Figure 6 biology-11-01049-f006:**
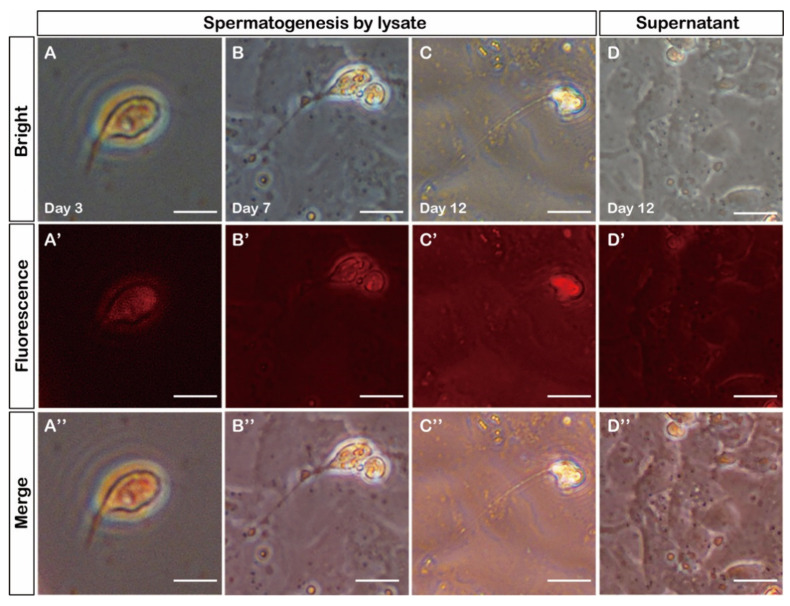
Induced differentiation *in vitro*. (**A**–**C**) SG3 cells were cultured with the lysate of CnGSC cells, and SG3 cells elongated to produce sperm cells. SG3 cells began to stretch out its tail at day 3; the head shrank and the tail pulled longer at day 12. (**D**) No sperm cells were observed after SG3 cells were cultured in supernatant for 12 days. (Bars = 10 µm in (**A**–**D”**)).

**Figure 7 biology-11-01049-f007:**
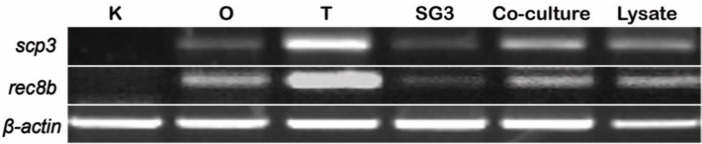
Meiotic gene expression *in vitro*. Expression of meiotic genes by RT-PCR with primers for *scp3* and *rec8b*. Expression of *scp3* and *rec8b* transcripts is low in undifferentiated SG3 cells. Elevated expression of both genes is observed in SG3 cells under induction culture conditions. K, head kidney; O, ovary; T, testis.

**Table 1 biology-11-01049-t001:** Primers used for reverse transcription–polymerase chain reactions (RT-PCR).

Gene		Primer Sequence	
Name	Species	Forward Primer	Reverse Primer
*dazl*	*C. nasus*	CTCGAGATGGATATCAACAAGCC	CAGCACAGTCAACATAGTC
*piwi*	*C. nasus*	CGACATCCACCAGCACAGA	AACGCCACGCATCTCCTT
*vasa*	*C. nasus*	CGCCATCTTCAATCAGTTCCA	AGTGTCTGCCTCTCCTCCT
*clu*	*C. nasus*	TCTCTGCTCTGTGTCTTATC	AACTTCTTGTGGTCCTCTC
*fshr*	*C. nasus*	GTGGTGCTGGTGTTGCTGCTTA	TGGACGAGTGAGTAGATAGTGCCTTC
*sox9b*	*C. nasus*	TGGACCCCTACCTGAAGATG	AGTCCAGTCGTAGCCCTTGA
*hsd3β*	*C. nasus*	GTGGTGGTGGTAGCGAAGT	GCCTCCGACAGCATACAGT
*actin*	*C. nasus*	TTCAACACCCCCGCCATGTAC	CCTCCGATCCAGACAGAGTATT
*scp3*	*Medaka*	GGAGCATCTGTGGAGCAACT	TCTGCAGCTTACATGGCCAA
*actin*	*Medaka*	TTCAACAGCCCTGCCATGTAC	CCTCCAATCCAGACAGAGTATT

## Data Availability

Not applicable.

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
