# Peer review of "Establishment of a Coilia nasus Gonadal Somatic Cell Line Capable of Sperm Induction In Vitro"

_biology, 2022, doi:10.3390/biology11071049_

Round 1

Reviewer 1 Report

In general, the present manuscript: “Establishment of a Coilia nasus gonadal somatic cell line capable of sperm induction in vitro” by Kan and colleagues focused to establish a gonadal somatic cell line (called CnGSC) capable of sperm induction, and it has an interesting goal. On the other hand, some specific comments are given bellow, to help improving the quality of the manuscript reviews.

Final remark: this current manuscript needs minor review.

- In general, I really appreciate this work with careful design, abundant data, and good text organization. In addition, it’s important to note that the manuscript is well structured and written (English grammar and composition). Congratulation.

- Usually, key words are words that do not contain in the title of the manuscript. Review the entire of manuscript.

- Currently, most of the scientific manuscript are presented as hypotheses to be more attractive and interesting than description of goals. The present manuscript can be presented with hypothesis. We suggest the authors to present this manuscript with more attractive hypothesis and to make the manuscript more interesting.

- Suggestion about rt-PCR analysis: Bustin et al. 2009: The MIQE Guidelines: Minimum Information for Publication of Quantitative Real-Time PCR Experiments. Clinical Chemistry, 55: 611-622. In this document, the authors find definitions and procedures used, such as the use of more than one endogenous gene and/or controls: for example, beta-actin gene levels and more one for internal control; reference genes for normalization (that difference of housekeeping genes). Present manuscript uses only one endogenous genes. Why?

- If possible, insert high quality and larger figures.

- some references need to be formatted according to the journal's authorguidence

Author Response

Dear Editors and Reviewers,

Thank you for giving me the opportunity to submit a revised draft of our manuscript. We appreciate the time and effort that you and the reviewers have dedicated to providing your valuable feedback on our manuscript. We are grateful to the reviewers for their insightful comments on our paper. Those comments are all valuable and very helpful for revising and improving our paper, as well as the important guidelines which are very helpful to make our research more efficient. We have been able to incorporate changes to reflect most of the suggestions provided by the reviewers. We have highlighted the changes within the manuscript. 

Here is a point-to-point response to the reviewers’ comments and concerns;

Reviewer #1: In general, the present manuscript: “Establishment of a Coilia nasus gonadal somatic cell line capable of sperm induction in vitro” by Kan and colleagues focused to establish a gonadal somatic cell line (called CnGSC) capable of sperm induction, and it has an interesting goal. On the other hand, some specific comments are given bellow, to help improving the quality of the manuscript reviews.

Final remark: this current manuscript needs minor review.

1.In general, I really appreciate this work with careful design, abundant data, and good text organization. In addition, it’s important to note that the manuscript is well structured and written (English grammar and composition). Congratulation.

Response: Thank you for praising our work. We would like to thank the referee again for taking the time to review our manuscript.

2.Usually, key words are words that do not contain in the title of the manuscript. Review the entire of manuscript.

Response: Thanks you for pointing this important point. According to your suggestion, we have reviewed the entire of manuscript and replaced words in keywords. And now keywords are “germplasm conservation; gonadal cell culture; spermatogonial stem cells; differentiation; spermatogenesis; spermatids”.

3.Currently, most of the scientific manuscript are presented as hypotheses to be more attractive and interesting than description of goals. The present manuscript can be presented with hypothesis. We suggest the authors to present this manuscript with more attractive hypothesis and to make the manuscript more interesting.

Response: Thanks for your careful reading of our paper and for providing us with some keen scientific insight. According to your suggestion, we have added an attractive hypothesis in the Introduction as follows: “This paper set out to address this research gap. Recently, we have cloned germ cell genes of C. nasus [26]. In the present study, we established a gonadal somatic cell line of C. nasus called CnGSC and investigated the somatic cells capable of sperm induction. It was hypothesized that spermatogenesis in vitro would be associated with the interactions between germ cells and somatic cells in fish.”.

4.Suggestion about rt-PCR analysis: Bustin et al. 2009: The MIQE Guidelines: Minimum Information for Publication of Quantitative Real-Time PCR Experiments. Clinical Chemistry, 55: 611-622. In this document, the authors find definitions and procedures used, such as the use of more than one endogenous gene and/or controls: for example, beta-actin gene levels and more one for internal control; reference genes for normalization (that difference of housekeeping genes). Present manuscript uses only one endogenous genes. Why?

Response: Thank you for your rigorous consideration. We agree with you that more one for internal control, however, RT-PCR analysis in our experiment is reverse transcription-PCR different from real-time PCR. And β-actin has been frequently used as an internal control (gene) or as a housekeeping gene to normalize the expression of the target gene(s) or mRNA levels between different samples (Li et al., 2010). Many studies have used β-actin as the internal control gene of RT-PCR (Poursaeid et al., 2020, Setthawong et al., 2019, Sun et al., 2015). Therefore, β-actin could be used as a reliable internal reference gene for real-time PCR based quantitation of gene expression studies.

Li, Z.; Yang, L.; Wang, J.; Shi, W.; Pawar, R.A.; Liu, Y.; Xu, C.; Cong, W.; Hu, Q.; Lu, T.; Xia, F.; Guo, W.; Zhao, M.; Zhang, Y. Beta-Actin is a useful internal control for tissue-specific gene expression studies using quantitative real-time PCR in the half-smooth tongue sole Cynoglossus semilaevis challenged with LPS or Vibrio anguillarum. Fish Shellfish Immunol. 2010, 29, 89-93.

Poursaeid, S.; Kalbassi, M.R.; Hassani, S.N.; Baharvand, H. Isolation, characterization, in vitro expansion and transplantation of Caspian trout (Salmo caspius) type a spermatogonia. Gen Comp Endocrinol. 2020, 289, 113341.

Setthawong, P.; Phakdeedindan, P.; Tiptanavattana, N.; Rungarunlert, S.; Techakumphu, M.; Tharasanit, T. Generation of porcine induced-pluripotent stem cells from Sertoli cells. Theriogenology. 2019, 127, 32-40.

Sun, A.; Chen, S.L.; Gao, F.T.; Li, H.L.; Liu, X.F.; Wang, N.; Sha, Z.X. Establishment and characterization of a gonad cell line from half-smooth tongue sole Cynoglossus semilaevis pseudomale. Fish Physiol Biochem. 2015, 41, 673-683.

5.If possible, insert high quality and larger figures.

Response: Thanks for your nice suggestion. We apologize if our figures did not show clearly. We have uploaded high quality pictures.

6.some references need to be formatted according to the journal's authorguidence

Response: Thanks for your careful reading. We have corrected the references format according to the journal's author guidelines.

Reviewer 2 Report

Dear Authors,

Please find below my suggestions.

--

Keywords

- Some words are in the Title. Please replace them.

1. Introduction

Line 50-51. "Research on fish germplasm resources conservation is increasingly important for the protection of endangered fish." Please use references as examples.

Line 57. Change to "Oncorhynchus mykiss (Walbaum 1792)". Please include all author of the species mentioned in text (e.g., Coilia nasus).

Line 65-66. Please use references as examples.

Line 72-74. Reference is needed.

2. Materials and Methods

Line 98. Should be Coilia nasus

Line 98- 105. Are there need of reference?

General comment. I believe methods should be better referenced.

3. Results

Well described.

4. Discussion

Line 286-298. Please left clear to the readers the novelty of your work.

Author Response

Reviewer #2: Please find below my suggestions.

1.Keywords

1.1Some words are in the Title. Please replace them.

Response: Thanks for your reminding us this important point. According to your suggestion, we have reviewed the entire of manuscript and replaced words in keywords. And now keywords are “germplasm conservation; gonadal cell culture; spermatogonial stem cells; differentiation; spermatogenesis; spermatids”.

2.Introduction

2.1Line 50-51. "Research on fish germplasm resources conservation is increasingly important for the protection of endangered fish." Please use references as examples.

Response: Thanks for your reminding us this important point. We have already added some references in the manuscript and the final sentence is “Research on fish germplasm resources conservation is increasingly important for the protection of endangered fish [2, 3].”.

[2] Goswami, M.; Mishra, A. Bio-banking: An Emerging Approach for Conservation of Fish Germplasm. Poultry, Fisheries & Wildlife Sciences. 2016, 4, 143.

[3] Kouba, A. J.; Lloyd, R. E.; Houck, M. L.; Silla, A. J.; Calatayud, N.; Trudeau, V. L.; Clulow, J.; Molinia, F.; Langhorne, C.; Vance, C.; Arregui, L.; Germano, J.; Lermen, D.; Della Togna, G. Emerging trends for biobanking amphibian genetic resources: The hope, reality and challenges for the next decade. Biological Conservation. 2013, 164, 10-21.

2.2Line 57. Change to "Oncorhynchus mykiss (Walbaum 1792)". Please include all author of the species mentioned in text (e.g., Coilia nasus).

Response: Thanks for your careful reading. We have added the Latin names of all species when it first appeared in the text. The name of species in Introduction are as follows: “half-smooth tongue sole (Cynoglossus semilaevis), yellow croaker (Larimichthys crocea), zebrafish (Danio rerio), medaka (Oryzias latipes)” and “mice (Mus musculus)” in discussion.

2.3Line 65-66. Please use references as examples.

Response: Thanks for your reminding us this important point. We have added the reference as following “Specifically, spermatogenesis in lower vertebrates can proceed in vitro, and sperm production in primary testicular culture has been reported for several fish species [11].”.

[11] Schulz, R. W.; de Franca, L. R.; Lareyre, J. J.; Le Gac, F.; Chiarini-Garcia, H.; Nobrega, R. H.; Miura, T. Spermatogenesis in fish. Gen Comp Endocrinol. 2010, 165, 390-411.

2.4Line 72-74. Reference is needed.

Response: Thanks for your careful reading. Because the sentences of lines 72-75 cited the same reference, the related reference was added at the end of the last sentence.

  1. Materials and Methods

3.1Line 98. Should be Coilia nasus

Response: Thank you for pointing this out. We have changed “C. nasus” into “Coilia nasus”.

3.2Line 98- 105. Are there need of reference?

Response: Thanks for your careful reading. We have added the reference in the manuscript that “Primary cell culture was performed as described by [27]”.

[27] Hong, Y.; Schartl, M. Isolation and differentiation of medaka embryonic stem cells. Methods Mol Biol. 2006, 329, 3-16.

3.3General comment. I believe methods should be better referenced.

Response: Thanks for your nice suggestion. We have added the references in the sections 2.2 and 2.5 that “Cryopreservation and recovery were performed as described previously with minor modifications [27]”, and “Phagocytosis assay was carried out accordingly to previously described [15]” respectively.

[27] Hong, Y.; Schartl, M. Isolation and differentiation of medaka embryonic stem cells. Methods Mol Biol. 2006, 329, 3-16.

[15] Sakai, N. Transmeiotic differentiation of zebrafish germ cells into functional sperm in culture. Development. 2002, 129, 3359-3365.

4.Results

4.1Well described.

Response: Thank you very much for your affirmation and approval to my article.

5.Discussion

5.1Line 286-298. Please left clear to the readers the novelty of your work.

Response: Thanks for your significant reminder. We have offered the novelty of our work that “Our study provides the first demonstration of long-term culture of endangered fish gonadal somatic cells in vitro. Several experimental evidences have confirmed the characteristics of the CnGSC cell line. First, CnGSC was of a gonadal origin, showing a fibroblast-like phenotype rather than round in culture [16], and was provided with typical phagocytic activity [35]. Second, it expressed the marker genes (clu, fshr, hsd3β, sox9b) of somatic cells but not germ cell marker genes (dazl, piwi, vasa). Third, and most important, it could induce spermatogonial stem cells to differentiate from meiosis into sperm. An enhanced expression level of both meiotic marker genes (rec8b, scp3) in the spermatogenesis system was consistent with the result of inducing mouse spermatocytes in vitro [40]. Thus, the first endangered fish gonadal cell line (CnGSC) appeared to be derived from the fish testis and characterized the role of gonadal somatic cells, and expanding existing knowledge.”.

In addition, the detailed specific changes are in the latest manuscript.

Acknowledgement: We appreciate for Editors and Reviewers’ warm work earnestly, and hope that the correction will meet with approval.

Once again, thank you very much for your comments and suggestions.
